# Isolation and Characterization of Feline Wharton’s Jelly-Derived Mesenchymal Stem Cells

**DOI:** 10.3390/vetsci8020024

**Published:** 2021-02-07

**Authors:** Min-Soo Seo, Kyung-Ku Kang, Se-Kyung Oh, Soo-Eun Sung, Kil-Soo Kim, Young-Sam Kwon, Sungho Yun

**Affiliations:** 1Laboratory Animal Center, Daegu-Gyeongbuk Medical Innovation Foundation, Daegu 41061, Korea; msseo@dgmif.re.kr (M.-S.S.); kangkk@dgmif.re.kr (K.-K.K.); osek1204@dgmif.re.kr (S.-K.O.); sesung@dgmif.re.kr (S.-E.S.); kslac@dgmif.re.kr (K.-S.K.); 2Department of Veterinary Toxicology, College of Veterinary Medicine, Kyungpook National University, Daegu 41566, Korea; 3Department of Veterinary Surgery, College of Veterinary Medicine, Kyungpook National University, Daegu 41566, Korea

**Keywords:** feline, Wharton’s jelly, mesenchymal stem cell, characterization

## Abstract

Wharton’s jelly is a well-known mesenchymal stem cell source in many species, including humans. However, there have been no reports confirming the presence of mesenchymal stem cells in Wharton’s jelly in cats. The purpose of this study was to isolate mesenchymal stem cells (MSCs) from the Wharton’s jelly of cats and to characterize stem cells. In this study, feline Wharton’s jelly-derived mesenchymal stem cells (fWJ-MSCs) were isolated and successfully cultured. fWJ-MSCs were maintained and the proliferative potential was measured by cumulative population doubling level (CPDL) test, scratch test, and colony forming unit (CFU) test. Stem cell marker, karyotyping and immunophenotyping analysis by flow cytometry showed that fWJ-MSCs possessed characteristic mesenchymal stem cell markers. To confirm the differentiation potential, we performed osteogenic, adipogenic and chondrogenic induction under each differentiation condition. fWJ-MSCs has the ability to differentiate into multiple lineages, including osteogenic, adipogenic and chondrogenic differentiation. This study shows that Wharton’s jelly of cat can be a good source of mesenchymal stem cells. In addition, fWJ-MSCs may be useful for stem cell-based therapeutic applications in feline medicine.

‡ Yun and Kwon contributed equally to this work as co-corresponding authors.

## 1. Introduction

Mesenchymal stem cells (MSCs), known as multipotent stem cells or mesenchymal stromal cells, have unique characteristics such as fibroblast-like morphology, expression of specific surface markers, and multipotent differentiation capacity [1,2]. MSCs were proposed as an important option in the field of regenerative medicine and immunotherapy due to their immunosuppressive properties, their ability to produce cytokines or growth factors, and their ability to differentiate multilineage such as osteogenic, chondrogenic, myogenic skeletal muscle tissue, among others [3].

MSCs have been identified in multiple tissue sources, such as adipose tissue, umbilical cord blood, placenta, skeletal muscle, skin, and bone marrow [4,5,6,7,8,9]. In addition, many studies in the veterinary field have reported the isolation of MSCs from various animals such as dogs, cats, horses, pigs, cows, and mice [10,11,12,13,14,15].

Studies have shown that feline MSC has therapeutic effects for gingivostomatitis, asthma, chronic kidney disease, and enteritis in feline medicine [16,17,18,19]. Feline MSC has been isolated from bone marrow, adipose tissue, and amniotic membrane [11,20,21]. However, there has been no report of MSCs in feline Wharton’s jelly. Wharton’s jelly is a gelatinous substance in the umbilical cord that contains fibroblast-like stromal cells [22]. Wharton’s jelly-derived MSC has been suggested to be a promising source for stem cells to be used in regenerative medicine and immunomodulatory therapy [23,24]. In this study, we have established feline Wharton’s jelly-derived stem cells (fWJ-MSCs) by characterizing mesenchymal stem cell potency, and it represents a good resource for stem cell therapy in feline medicine.

## 2. Materials and Methods

### 2.1. Animals

The umbilical cord was obtained using pregnant cats (*n* = 2, 4~5 years, mixed breed) who visited Kyungpook National University Animal Hospital. Applicable institutional and governmental regulations concerning the ethical use of animals were followed during the course of this research. In the cesarean-section delivery, the cats were pre-medicated with 0.1 mg/kg acepromazine maleate (Samwoo Medical, Yesan, Korea), and then 5 mg/kg propofol (Myeongmun Pharmaceutical, Seoul, Korea) was injected intravenously to induce anesthesia. Isoflurane (Hana Pharmaceutical, Hwasung, Korea) was used to maintain anesthesia. Under sterile conditions, umbilical cords were collected from cesarean section deliveries. Cats were hospitalized and discharged two weeks after surgery.

### 2.2. Cell Isolation

Cell isolation was performed through modifications such as those previously described [1,2,25]. The collected tissue was delivered into the sterile specimen cup and then washed with 0.9% physiological saline to remove as much blood as possible. Blood vessels were mechanically removed with forceps. After that, the feline Wharton’s jelly tissue was finely minced, using a surgical blade, and resuspended in 2 mg/mL collagenase type I solution (Sigma-Aldrich, St. Louis, MO, USA) at 37 °C for approximately 3 h. After enzyme digestion, it was washed in phosphate-buffered saline (PBS) (Thermo Fisher Scientific, Waltham, MA, USA) by centrifugation at 1811× *g* for 5 min. Cell pellets were resuspended in the basal culture medium, low glucose Dulbecco’s modified Eagle’s medium (LG-DMEM; Thermo Fisher Scientific, Waltham, MA, USA) containing 10% FBS (fetal bovine serum; Thermo Fisher Scientific, Waltham, MA, USA). The cells were seeded into T-75 polystyrene cell culture flasks (Corning, Corning, NY, USA) and incubated in a humidified atmosphere with 5% CO_2_. Culture medium was changed 3 times a week and passaged once the cells reached 80–90% confluency.

### 2.3. Reverse Transcriptase Polymerase Chain Reaction

Total RNA was extracted from the cultured cells at passage 5 with an RNeasy Mini Kit (Qiagen, Hilden, Germany) according to the manufacturer’s protocol. RNA concentrations were measured by Nanodrop 2000 (Thermo Fisher Scientific, Waltham, MA, USA). cDNA was prepared by 1 mg of total RNA for reverse transcription using Superscript II reverse transcriptase (Thermo Fisher Scientific, Waltham, MA, USA) and oligo dT primers (Thermo Fisher Scientific, Waltham, MA, USA). The cDNA was amplified using T100™ Thermal Cycler (Biorad, Hercules, CA, USA). The PCR primers are shown in the Table 1. PCR products were visualized with ethidium bromide on a 3% agarose gel.

### 2.4. Cumulative Population Doubling Level Analysis

The proliferation and growth efficiency of fWJ-MSCs were determined by cumulative population doubling level (CPDL) assay, using the formula CPDL = ln (N_f_/N_i_)ln_2_, where N_i_ is the initial seeding cell number, N_f_ is the final harvest cell numbers, and ln is the natural log. fWJ-MSCs (5 × 10^4^ cells) were plated in triplicate in a 6-well culture plate (Corning, Corning, NY, USA) and subcultured 4 days later. The final cell numbers were counted, and 5 × 10^4^ cells were replated. To calculate the cumulated doubling level, a population doubling for each passage was calculated and then added to the population doubling levels of the previous passages from passage 5 to 16.

### 2.5. Scratch Test, Colony Forming Unit Test

In the scratch test, fWJ-MSCs were incubated in 6-well plates until 90% confluence was reached. The original medium was removed, and the cells washed with PBS. Cell scratches were made with a 200 μL pipette tip, and residual cells were washed with PBS. DMEM culture medium containing 5% FBS was used for cell culture for 24 h. After 24 h, inverted microscope was used to observe the healing state of the cells.

In the CFU test, Passage 5 fWJ-MSCs were seeded at 1 × 10^3^ cells in 6-well plates and incubated in DMEM culture medium containing 10% FBS. After 2 weeks, the plastic adherent colonies were stained with 1% toluidine blue (Scytek, Logan, UT, USA).

### 2.6. Karyotype Analysis

In order to detect any chromosomal abnormalities in the fWJ-MSCs, karyotyping was conducted followed by the standard methods at passage 5. Briefly, fWJ-MSCs were maintained with 500 μL colcemid (Thermo Fisher Scientific, Waltham, MA, USA) in the incubator (37 °C, 5% CO_2_) for 1 h. The cells were pelleted at 201 g for 10 min and were suspended in a hypotonic solution (0.075 M KCl; Sigma-Aldrich, St. Louis, MO, USA) and incubated for 20 min at 37 °C; then, the cells were fixed by washing three times in Canoy’s fixative (methanol: glacial acetic acid = 3:1; Sigma-Aldrich, St. Louis, MO, USA). Chromosome spreads were obtained by pipetting suspension drops onto clean glass and allowing them to air dry. Cells undergoing metaphase were captured with a charge-coupled device camera (Olympus, Tokyo, Japan), chromosomes were counted, and the banding pattern was analyzed by ChlPS-Karyo software (GenDix Inc., Seoul, Korea).

### 2.7. Flow Cytometry

To determine the immunophenotype of fWJ-MSCs, cells were stained with specific antibodies for fluorescence-activated cell sorter (FACS) analysis, following the protocol provided by the supplier (Beckman Coulter, Brea, CA, USA). Briefly, the fWJ-MSCs were trypsinized and washed several times with PBS at passage 5. The suspended cells were aliquoted (approximately 1 × 10^6^ cells) for specific antibody staining. The cells were immunostained with the antibodies shown in the Table 2. The antibodies were conjugated with fluorescein isothiocyanate (FITC) or phycoerythrin (PE). Analysis was determined by the use of FACS (Gallios Flow Cytometer; Beckman Coulter, Brea, CA, USA) and software (Kaluza for Gallios; Beckman Coulter, Brea, CA, USA).

### 2.8. Osteogenic Differentiation

Osteogenic differentiation medium (StemPro Osteogenesis Differentiation Kit; Thermo Fisher Scientific, Waltham, MA, USA) was used to determine osteogenic differentiation capability. At passage 5, when cells reached confluency of 80 to 90%, the medium was changed to the osteogenic differentiation medium, and incubated for 3 weeks, changing once every 3 days. For comparison, passage 5 cells were used as undifferentiated cells. After 3 weeks, the calcium deposition was detected by staining with Alizarin Red S and Von Kossa. For Alizarin Red S staining, the cells were washed with PBS and fixation was performed with 70% ethanol for 1 h at 4 °C. The cells were then washed several times with distilled water. The cells were stained with Alizarin Red S (IHCworld, Woodstock, MD, USA) for 10 min at room temperature. The cells were washed of non-specific dye with five washes of distilled water. For quantitative measurements, Alizarin Red S staining was solubilized using 100 mM cetylpyridinium chloride (Sigma-Aldrich, St. Louis, MO, USA) for 1 h. Solubilized Alizarin Red S absorbance was measured at 570 nm using a spectrophotometer. For Von Kossa staining, the cells were stained with 5% silver nitrate (Scytek, Logan, UT, USA) for 30 to 60 min with exposure to ultraviolet light, followed by 5% sodium thiosulfate (Scytek, Logan, UT, USA) for 2~3 min, and then counterstained with Nuclear Red stain (Scytek, Logan, UT, USA) for 5 min.

### 2.9. Adipogenic Differentiation

To determine if the fWJ-MSCs could differentiate into adipocytes, the cells were treated with adipogenic differentiation media (StemPro Adipogenesis Differentiation Kit; Thermo Fisher Scientific, Waltham, MA, USA, USA). At passage 5, when cells reached confluency of 80 to 90%, the medium was changed to the adipogenic differentiation medium, and the cells were incubated for 3 weeks, changing media once every 3 days. For comparison, passage 5 cells were used as undifferentiated cells. After 3 weeks, Oil Red O staining (Scytek, Logan, UT, USA) was performed to detect lipid droplets. Cells were fixed with 10% neutral buffered formalin for fixation at least 1 h and rinsed with 60% isopropanol prior to incubation in freshly diluted Oil Red O for 10 min. Stains were solubilized using 100% isopropanol, and the resulting absorbance was measured at 500 nm using a spectrophotometer.

### 2.10. Chondrogenic Differentiation

To confirm that the fWJ-MSCs did differentiate into chondrocytes, chondrogenic differentiation medium (StemPro Chondrogenesis Differentiation Kit; Thermo Fisher Scientific, Waltham, MA, USA) was used. First, cells (5 × 10^5^ cells) were seeded in a 15 mL polypropylene tube and centrifuged to a pellet at passage 5. Pellets were cultured in 1 mL of chondrogenic differentiation medium and incubated for 4 weeks. Differentiation medium was changed 3 times a week. After differentiation, pellets were embedded in paraffin and 4 μm sections were cut. For histological evaluation, the sections were stained with toluidine blue. A cell pellet slice mounted on a slide was deparaffinized and hydrated with distilled water. For toluidine blue staining, the slide was immersed in toluidine blue working solution (Scytek, Logan, UT, USA) for 1 min. Excess unbound stain was removed by several washes using distilled water. The slide was quickly dehydrated with sequential washes of 95% and absolute alcohols and cleared in xylene and covered with balsam solution and a cover slip. Secondly, at passage 5 when cells reached a confluency of 80 to 90%, the medium was changed to the chondrogenic differentiation medium, and incubated for 3 weeks, changing once every 3 days. For comparison, passage 5 cells were used as undifferentiated cells. After 3 weeks, Toluidine blue staining (Scytek, Logan, UT, USA) was performed to detect chondroid. For quantitative measurements, toluidine blue staining was solubilized using 100 mM cetylpyridinium chloride (Sigma-Aldrich, St. Louis, MO, USA) for 1 h. Solubilized toluidine blue absorbance was measured at 600 nm using a spectrophotometer.

### 2.11. Quantitative Real-Time Polymerase Chain Reaction

Quantitative real-time PCR (qRT-PCR) was performed by mixing cDNA with primers and LightCycler^®^ 480 SYBR Green I Master (Roche Diagnostics, Risch-Rotkreuz, Switzerland). qRT-PCR was performed using a LightCycler 480 II with supplied software (Roche Applied Science, Penzberg, Germany) according to the manufacturer’s instructions. RNA expression levels were compared after normalization to endogenous glyceraldehyde-3-phosphate dehydrogenase (*GAPDH*). The primer sequences used in this study are listed in Table 3.

### 2.12. Statistical Analysis

The data were analyzed by Student’s *t*-test using Excel software (Microsoft, Redmond, WA, USA) and expressed as the mean ± standard error. Statistically significant data are indicated by asterisks (*** *p* < 0.001, ** *p* < 0.01, * *p* < 0.05).

## 3. Results

### 3.1. Isolation and Culture of fWJ-MSCs

Feline Wharton’s jelly was obtained after cesarean section, and collected samples were moved to a sterile culture dish (Figure 1A). We isolated fWJ-MSCs from the Wharton’s jelly using conventional methods for stem cell isolation and culture, as previously described [25]. Cellular morphology was fibroblast-like and spindle-like in identity (Figure 1B). To determine the self-renewal capacity of fWJ-MSCs, we measured and calculated the proliferation via cumulative population doubling level (CPDL). A consistent increasing rate of cell proliferation through the cumulative population was observed (Figure 1C).

### 3.2. Scratch Test, Colony Forming Unit (CFU) Test

A scratch test was performed to investigate the migration ability of fWJ-MSCs. Images taken at different time lapses after 24 h of scratch showed the possibility of migration of fWJ-MSC (Figure 1D). A CFU test demonstrated that fWJ-MSCs can generate new colonies from a single cell and establish multiple new colonies (Figure 1E).

### 3.3. Expression Pattern of Stem Cell Markers

To measure gene expression levels of stem cell markers, reverse transcriptase polymerase chain reaction (RT-PCR) was performed in passage 5. Stem cell markers such as *OCT4*, *SOX2*, *KLF4* and *MYC* showed expression patterns as a function of stemness. (Figure 1F).

### 3.4. Analysis of Karyotype

We performed karyotyping using fWJ-MSC to identify normal chromosome numbers using our culture process. Cells were confirmed to contain 38 chromosomes, which is the expected normal karyotype of felines (Figure 2A).

### 3.5. Flow Cytometry

To confirm the immunophenotype of fWJ-MSCs, cell surface specific markers were examined by fluorescence-activated cell sorter (FACS) analysis. As a result of FACS analysis, fWJ-MSCs have an expression pattern consistent with the MSC immunophenotype (Figure 2B). The results showed that fWJ-MSCs showed positive expression of *CD44* (100%), *CD90* (65.96%) and *CD105* (60.62%), which are well-known typical MSCs markers. However, cells were negative for the expression of immune cells markers (*CD14*; 10.16%) and hematopoietic cells markers (*CD34*; 11.26%, *CD45*; 9.04%).

### 3.6. Induction of Differentiation

To evaluate the differentiation ability of fWJ-MSCs, three different types of differentiation assays were conducted. Under the osteogenic differentiation conditions, the cells were strongly positively stained with Alizarin Red S (Figure 3A–C) and Von Kossa staining (Figure 3D,E). Not surprisingly, under the basal culture condition, the cells were not stained with Alizarin Red S (Figure 3G,H) and Von Kossa staining (Figure 3I,J). Compared with the basal culture condition, osteogenic differentiated cells showed approximately 67-fold greater values of absorbance in Alizarin Red S staining (Figure 3K). After differentiation, the gene expression levels of *MSX2*, *COL1A1* were increased compared to controls (Figure 3L,M).

Under adipogenic differentiation conditions, fatty droplets were observed via both unstained (Figure 4A) and positive Oil Red O staining (Figure 4B,C). The basal culture medium was used as undifferentiated cells, which ultimately displayed negative staining (Figure 4E,F). The differentiated cells showed approximately three-fold higher absorbance than that of the undifferentiated cells (Figure 4G). After adipogenic differentiation, gene expression levels of adipogenic-associated markers such as *LPL*, *LEPTIN* and *FABP4* were increased in treated cells compared to undifferentiated cells (Figure 4H–J).

Under the chondrogenic differentiation conditions in six-well plates, the cells were positively stained with toluidine blue and undifferentiated cells were not stained (Figure 5A–F). Chondrogenic differentiated cells showed approximately three-fold greater values of absorbance in toluidine blue staining compared with the undifferentiated cells (Figure 5G). After chondrogenic differentiation in tubes, we examined the pellet formation in the bottom of the polypropylene tube. The pellet appeared to be ovoid and opaque (Figure 5H). The pellet was then fixed and stained with toluidine blue (Figure 5I), which exhibited positive staining patterns. *COL2A1* gene expression levels were increased under differentiation conditions, compared to undifferentiated conditions (Figure 5J).

## 4. Discussion

Feline MSCs were first isolated and characterized in the bone marrow [11], and subsequently feline MSCs have been derived from adipose tissue [20,26,27,28], amniotic membrane, and amniotic fluid [21,29]. In these studies, feline MSCs were characterized by morphological features, specific cell surface markers, and their capacity for differentiation. Wharton’s jelly is easy to obtain and a rich source of stem cells in various species, including humans [24,30]. However, little is known about stem cells from Wharton’s jelly in cats. Although the existence of MSCs in Wharton’s jelly has been demonstrated in many species, including humans, canines, swines, bovines, equines, chickens, and mice [25,31,32,33,34,35], our study is the first to demonstrate the isolation of fWJ-MSCs exhibiting morphology, proliferation, karyotyping, surface markers, and the capacity to differentiate into multi-lineage.

A previous study showed that Wharton’s jelly-derived stem cells (WJ-MSCs) have higher proliferative potential than MSCs from other sources such as bone marrow, adipose tissue, placenta, and amniotic membrane [36,37,38,39,40]. Additionally, WJ-MSCs have primitive features, because they exhibit several characteristics of embryonic stem cells (ESC), such as ESC-like antigens *Tra-1-60*, *Tra-1-81*, *SSEA-1* and *SSEA-4* [41], and numerous pluripotency genes including *OCT4* and *SOX2* [42]. WJ-MSCs are good candidates for therapeutic applications due to the ease of accessibility, lack of painful procedures, greater proliferative potential, less risk of contamination, and hypoimmunogenicity [24,43,44]. Another advantage of WJ-MSCs is that they can be cryopreserved after the birth of a baby and used for future purposes using biobanks [45].

In our study, we isolated and cultured cells from feline Wharton’s jelly. Cellular morphology of fWJ-MSCs was fibroblast-like and spindle-like, similar to other MSCs such as adipose-derived MSCs, bone marrow-derived MSCs, peripheral blood-derived MSCs, and umbilical cord blood-derived MSCs [20,25,33,46]. Additionally, fWJ-MSCs had similar proliferation and characteristics to MSCs derived from other feline tissues [29,47]. According to the International Society of Cellular Therapy, MSCs have positive expression of *CD44*, *CD90* and *CD105*, but negative expression of *CD14*, *CD34* and *CD45* surface antigens [48]. For FACS analysis of fWJ-MSCs, we used six CD markers (*CD14*, *CD34*, *CD44*, *CD45*, *CD90* and *CD105*) to distinguish the MSC phenotype. fWJ-MSCs were positive by flow cytometry for *CD44*, *CD90* and *CD105* and negative for *CD14*, *CD34* and *CD105*, similar to MSCs from other feline tissues and WJ-MSCs from other species [20,49]. fWJ-MSCs also had the potential for multiple lineage differentiation, including osteogenic, adipogenic and chondrogenic, similar to other species’ WJ-MSCs [25,49]. Based on this evidence, we suggest that cells derived from feline Wharton’s jelly in this study were MSCs. Limitedly, we have identified the minimum genes required to establish fWJ-MSCs, but further investigation into large panels of genes, such as RNA sequencing, may be required to clarify the origin of the cells. It is also necessary to compare large genetic panels between feline MSCs and MSCs of other species.

There are several reports using feline stem cells in therapeutic studies in cats. The stem cells used in most cases have been shown to originate from feline adipose tissue [16,17,18,19,50,51]. With regard to potential therapeutic uses, there is a need for a source of stem cells of established diversity.

## 5. Conclusions

These results show that we successfully isolated stem cells from feline Wharton’s jelly, which have both the properties of MSCs and the ability for self-renewal. Therefore, fWJ-MSCs could potentially be used for stem cell therapy in feline medicine. In addition, we suggest that fWJ-MSCs could be a potential source of stem cells, and thus could be useful for veterinary medicine.

## Figures and Tables

**Figure 1 vetsci-08-00024-f001:**
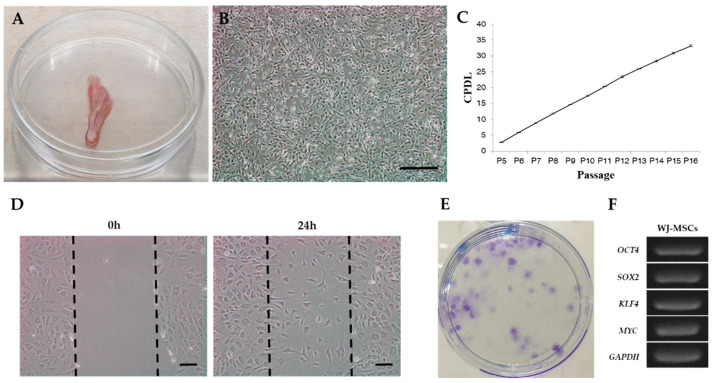
Primary culture of fWJ-MSCs and identification of cumulative population doubling level (CPDL). (**A**) Extracted feline umbilical cord tissue. (**B**) Image of isolated fWJ-MSCs. Cells exhibited a spindle shape, similar to other mesenchymal stem cells. Scale bar: 100 μm. (**C**) Cumulative growth curve of fWJ-MSCs. CPDL was measured from passage 5 to 16. (**D**) Scratch test. Figure shows representative images of fWJ-MSCs migration post-scratch assay obtained after 24 h. Scale bar: 10 μm. (**E**) Colony forming unit (CFU) test. At 2 weeks of culture, the CFUs were stained with toluidine blue to visualize the colonies generated. (**F**) RT-PCR. *OCT4*, *SOX2*, *KLF4*, *MYC* expression determined stemness.

**Figure 2 vetsci-08-00024-f002:**
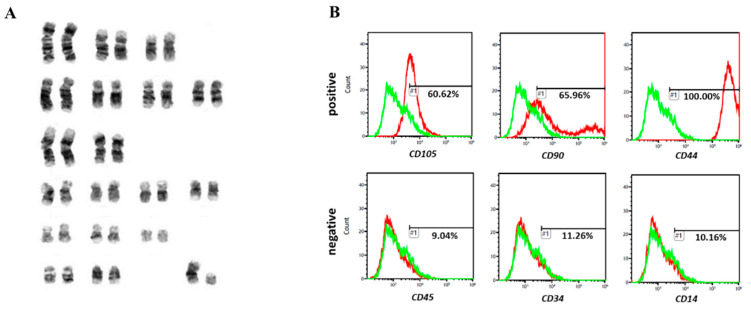
Karyotyping and flow cytometry analysis. (**A**) Karyotype of fWJ-MSCs at passage 5 showing a euploid number of chromosomes. (**B**) FACS analysis was performed at passage 5. Values show the intensity of the indicated antigen.

**Figure 3 vetsci-08-00024-f003:**
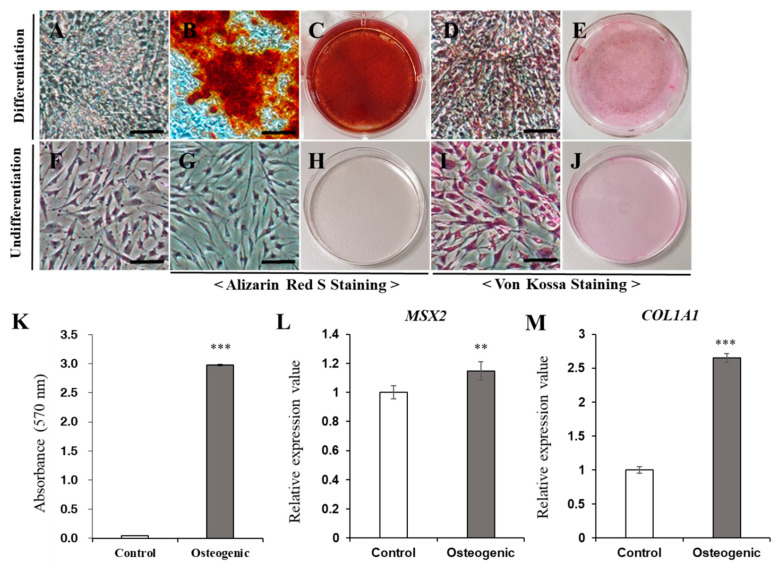
Osteogenic differentiation. (**A**–**J**) Osteogenic differentiation of fWJ-MSCs. (**A**,**F**) Bright field images of cells in osteogenic condition and control condition. (**B**–**E**,**G**–**J**) Alizarin Red S and Von Kossa staining after 3 weeks of osteogenic induction and control condition. Osteogenic differentiated cells (**A**–**E**) were grown in osteogenic induction medium. Negative control cells (**F**–**J**) were grown in low glucose DMEM with 10% FBS. Differentiated cells stained strongly with Alizarin Red S (**B**,**C**) and Von Kossa (**D**,**E**). Scale bar: 25 μm. (**K**) For quantification, Alizarin Red S-stained cells were solubilized with 100 mM cetylpyridinium chloride, and the absorbance was measured by spectrophotometer at 570 nm for 0.5 s. Compared with the negative control, differentiated cells showed 67-fold greater absorbance values. (**L**,**M**) qRT-PCR for detection of mRNA expression level of osteogenic-specific markers*: MSX2*, *COL1A1*, and *GAPDH* were used as references for evaluating the quality of mRNA. (Control, undifferentiated fWJ-MSC). Means ± standard deviations are plotted (*** *p* < 0.001), ** *p* < 0.01).

**Figure 4 vetsci-08-00024-f004:**
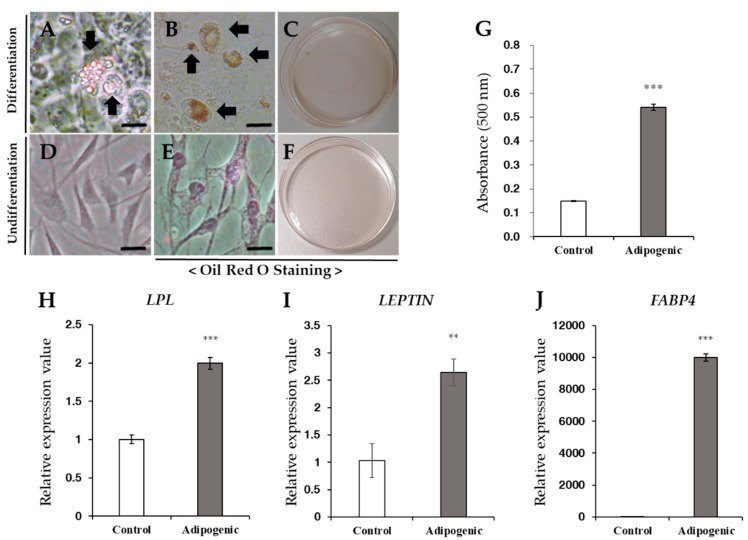
Adipogenic differentiation. (**A**–**F**) Adipogenic differentiation of fWJ-MSCs. Oil Red O staining was conducted after 3 weeks of adipogenic induction. (**A**–**C**) Adipogenic differentiated cells were grown in adipogenic induction media. Fatty droplets were strongly stained with Oil Red O (arrow: fatty droplet). (**D**–**F**) Negative controls showed no staining with Oil Red O. Scale bar: 10 μm. (**G**) For quantification, stained cells were solubilized with 100% isopropanol, and the absorbance was measured spectrophotometrically at 500 nm for 0.5 s. Compared with the negative control, differentiated cells showed 3-fold greater absorbance values. (**H**–**J**) qRT-PCR for detection of mRNA expression level of adipogenic-specific markers: *LPL*, *LEPTIN* and *FABP4*. *GAPDH* was used as the reference for evaluating the quality of mRNA. (Control, undifferentiated fWJ-MSC). Means ± standard deviations are plotted (*** *p* < 0.001), ** *p* < 0.01).

**Figure 5 vetsci-08-00024-f005:**
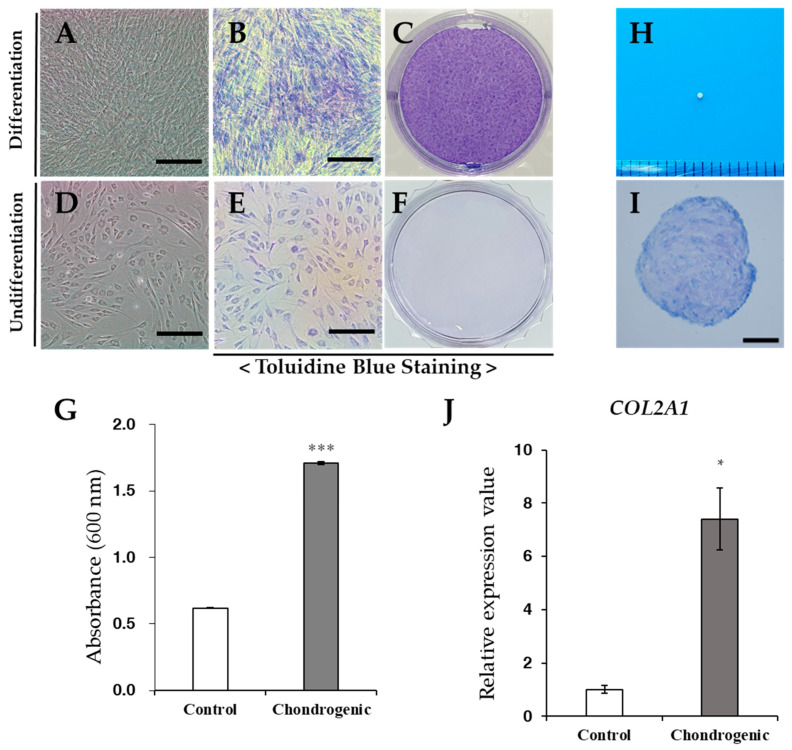
Chondrogenic differentiation. (**A**–**F**) Chondrogenic differentiation of fWJ-MSCs. Toluidine blue scheme after 3 weeks of chondrogenic induction. Scale bar: 50 μm. (**G**) qRT-PCR for detection of mRNA expression level of chondrogenic-specific markers: COL2A1. GAPDH was used as a reference for evaluating the quality of mRNA. (**H**) The shape of chondrogenic pellet. (**I**) Toluidine blue staining for chondrogenic pellet. Scale bar: 100 µm. (**J**) qRT-PCR for detection of mRNA expression level of osteogenic-specific markers: *COL2A1*. *GAPDH* was used as a reference for evaluating the quality of mRNA. (Control, undifferentiated fWJ-MSC). Means ± standard deviations are plotted (*** *p* < 0.001), * *p* < 0.05).

**Table 1 vetsci-08-00024-t001:** List of RT-PCR primers.

Genes	Forward Primer (5′-3′)	Reverse Primer (5′-3′)	Product Size
*OCT4*	GCCCGAAAGAGAAAGCGAAC	CGACGATTGCAGAACCACAC	161 bp
*SOX2*	GCCCTGCAGTACAACTCCAT	TGGAGTGGGAGGAAGAGGTA	175 bp
*KLF4*	ACCAAGAGCTCATGCCACCT	AAGGCTTCTCACCTGTGTGG	183 bp
*MYC*	AGGAGAAACGAGCTGAAACG	GTTCTCGTCGCTTCCTCAAC	181 bp
*GAPDH*	GTGGAGGGACTCATGACCAC	GTGAGCTTCCCATTCAGCTC	176 bp

**Table 2 vetsci-08-00024-t002:** List of fluorescence-activated cell sorter (FACS) antibodies.

Marker	Antibody	Company/Catalog number
*CD105*	MOUSE ANTI HUMAN *CD105*	BIO-RAD/MCA1557
*CD90*	PE Mouse Anti-Human *CD90*	BD Pharmingen/555596
*CD45*	FITC Mouse Anti-Human *CD45*	BD Pharmingen/555482
*CD44*	PE anti-mouse/human *CD44* Antibody	BioLegend/103024
*CD34*	FITC Mouse Anti-Human *CD34*	BD Pharmingen/555821
*CD14*	MOUSE ANTI HUMAN *CD14*	BIO-RAD/MCA1568

**Table 3 vetsci-08-00024-t003:** List of real-time PCR primers.

Genes	Forward Primer (5′-3′)	Reverse Primer (5′-3′)	Product Size
*MSX2*	GCCTCCAAGACACATGAGC	CCTGGGTCTCTGTGAGGTTC	185 bp
*COL1A1*	GAGCGGAGAATACTGGATCG	ATGCTCTCGCCATACCAGAC	180 bp
*LPL*	TGGCGGAGGAATTTCACTAT	AGGAGAAAGGCGACTTGGAG	176 bp
*LEPTIN*	AGCAGCTTGGCTGACAATTT	CCAGCAATCACTCCTGGTCT	178 bp
*FABP4*	CATCAGTGTGAATGGGGATG	CCACTTCTGCACCTGTACCA	169 bp
*COL2A1*	CCCTAGAGGTCCTCCTGGTC	CAAAGGCAGACATGTCGATG	188 bp
*GAPDH*	GTGGAGGGACTCATGACCAC	GTGAGCTTCCCATTCAGCTC	176 bp

## Data Availability

The data presented in this study are available on reasonable request from the corresponding author.

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
