# Peer review of "Isolation and Characterization of Feline Wharton’s Jelly-Derived Mesenchymal Stem Cells"

_vetsci, 2021, doi:10.3390/vetsci8020024_

Round 1
Reviewer 1 Report
In this article, entitled “Isolation and Characterization of Feline Wharton’s Jelly-Derived Mesenchymal Stem Cells”, the authors isolated and characterized mesenchymal stem cells (MSCs) from the feline Wharton’s jelly. In general, this article demonstrates a new source for the MSC and provides possibilities for therapeutic applications. I would like to suggest a few suggestions.
1. In Figure 1f, the authors need to compare the expression level of stemness genes with control (ex. Fibroblasts as a negative control or other stem cells as a positive control.
2. In Figure 3L, the expression level of the MSX2 from the osteogenic differentiation is not much higher than control although it is statistically significant. The authors need to check other markers for osteogenic differentiation such as Col1a1, Col1a2, and VDR.
Author Response
In this article, entitled “Isolation and Characterization of Feline Wharton’s Jelly-Derived Mesenchymal Stem Cells”, the authors isolated and characterized mesenchymal stem cells (MSCs) from the feline Wharton’s jelly. In general, this article demonstrates a new source for the MSC and provides possibilities for therapeutic applications. I would like to suggest a few suggestions.
- In Figure 1f, the authors need to compare the expression level of stemness genes with control (ex. Fibroblasts as a negative control or other stem cells as a positive control.
â–¶ Thank you for your kindly comments. I agree with the reviewer's comment. It is necessary to compare the expression level of stemness gene with the control, but it is difficult to obtain feline fibroblast commercially (There is no primary feline fibroblast cell line in ATCC.) and it is difficult to do it within the review period (7 days). Figure 1f is the result of confirming the presence of stemness gene expression rather than to see the expression level. If there is a chance to acquire feline fibroblast, we will plan further research by actively accepting reviewer's comment.
- In Figure 3L, the expression level of the MSX2 from the osteogenic differentiation is not much higher than control although it is statistically significant. The authors need to check other markers for osteogenic differentiation such as Col1a1, Col1a2, and VDR.
â–¶ Thank you for your kindly comments. As your comments, additional experiment was performed to confirm the level of the Col1a1 gene, and the results were added (noted by blue color; p5 table 3, p7 figure 3M, line 246-247).
Reviewer 2 Report
The authors aimed to established feline wharton’s jelly-derived stem cells (fWJ-MSC) by characterizing mesenchymal stem cell potency.
These results provide novel preclinical evidence that supports the use of fWJ-MSC as potential resource for stem cell therapy in feline medicine.
The study is easy to follow and covers an hot topic, but minor issues should be improved before publication. Several typos should be corrected thorough the text.
Conclusion Section: This paragraph required a general revision to eliminate redundant sentences and to add some "take-home message".
Author Response
The authors aimed to established feline wharton’s jelly-derived stem cells (fWJ-MSC) by characterizing mesenchymal stem cell potency.
These results provide novel preclinical evidence that supports the use of fWJ-MSC as potential resource for stem cell therapy in feline medicine.
The study is easy to follow and covers an hot topic, but minor issues should be improved before publication. Several typos should be corrected thorough the text.
Conclusion Section: This paragraph required a general revision to eliminate redundant sentences and to add some "take-home message".
â–¶ Thank you for your kindly comments. As your comments, we corrected the conclusion section that was noted by yellow color (p10 327-331).
Reviewer 3 Report
In this manuscript, Seo and Kang et al performed an impressive set of experiments to first confirm the presence of mesenchymal stem cells (MSCs) in Wharton’s jelly and then characterized these cells to show their stem cell potential. This is a very nice study and should have long-term implications for stem cell dependent therapeutic applications in feline medicine. However, I think the the authors have used only a few handful genes (OCT4, SOX2, KLF4 and MYC) to check the stem-ness of these cells using gene expression (qPCR). In theory, a large panel of genes should be tested by RNA-seq to unambiguously validate the origin of these cells. This type of analysis will also allow authors to compare the MSCs derived from cats versus other species. This could be a tremendous usefulness of this study. Hence, if the other reviews for this manuscript are favorable, then I would like the authors at least mention about this point as an immediate future goal in the discussion. I think this would make the implication of this study better and make a nice addition to the journal Veterinary Sciences.
Author Response
In this manuscript, Seo and Kang et al performed an impressive set of experiments to first confirm the presence of mesenchymal stem cells (MSCs) in Wharton’s jelly and then characterized these cells to show their stem cell potential. This is a very nice study and should have long-term implications for stem cell dependent therapeutic applications in feline medicine. However, I think the the authors have used only a few handful genes (OCT4, SOX2, KLF4 and MYC) to check the stem-ness of these cells using gene expression (qPCR). In theory, a large panel of genes should be tested by RNA-seq to unambiguously validate the origin of these cells. This type of analysis will also allow authors to compare the MSCs derived from cats versus other species. This could be a tremendous usefulness of this study. Hence, if the other reviews for this manuscript are favorable, then I would like the authors at least mention about this point as an immediate future goal in the discussion. I think this would make the implication of this study better and make a nice addition to the journal Veterinary Sciences.
â–¶ Thank you for your kindly comments. I agree with the reviewer's comment. As your comments, we added the discussion that was noted by green color (p10 318-321).

Reviewer 4 Report
The paper of Seo et al. is interesting because study an interesting stem cells source in the feline species.
The paper is well written and well organized with important experimental studies to define the characteristics of these stem cells. I have only some suggestions:
To define the characteristics of a stem cell line, it is very important to check the modification of these cells during the different passages in vitro. So, it would have been important to check the positivity to the mesenchymal markers and the negativity to the hematopoietic markers at different passages: for example, at passage 0, 5 and 10 ....
Moreover, I do not understand why the flow cytometry and differentiation studies were done at the step 5 and the gene expression studies through PCR at the step 3.
At last, we know that this source is heterologous so, in the case of therapy these cells will be used cryopreserved, so it would have been important to understand if these cells tolerate the cryopreservation and if after cryopreservation they maintain the same proliferative, differentiative properties and the expression of stem cell markers.
This paper need to some improvement about these issues
In addition, there are some suggestions:
Line 33-35 pag 1: please change “MSCs were proposed as an important option in the field of regenerative medicine and immunotherapy due to its immunosuppressive properties, its ability to produce cytokines or growth factors, and its ability to differentiate multilineage such…. “ In “MSCs were proposed as an important option in the field of regenerative medicine and immunotherapy due to their immunosuppressive properties, their ability to produce cytokines or growth factors, and their ability to differentiate multilineage such…”
Line 41 pag 1 “… cow, mouse (10-15).”Here are reported only citations about the isolation from BM but there are many papers in veterinary field about the isolation of cells from other sources as for example, amniotic membrane or amnioric fluid or wharthon jelly....please insert some papers about this (amniotic membrane, amniotic fluid and wharthon jelly were studied in many animal species by Lange-Consiglio et al.).
Line 72 pag 2: “by centrifugation at 3,000 rpm for 5 min.” please change with “g” that is an universal value
Line 80 pag 2: “Total RNA was….” Please insert here at what passage you have performed this study
Line 96 pag 3: “…levels of the previous passages….” Please insert here until to which passage you have performed this study
Line 110 pag 3: “…at 1,000 rpm for 10…” please insert “g” and not rpm
Line 11 pag 3: “….at 37°C. And the cells…” please change in “…at 37°C, then the cells..."
Line 129 pag 4: “Osteogenic differentiation medium…..” please, for each differentiation study insert that you have performed the negative control
Line 290 pag 9: “In previous study showed that…” please change in “Previous study showed that….”
Line 295 pag 9: “WJ-MSCs…” please, do not use the acronym at the at the beginning of a sentence. Please check through all the manuscript
Line 320 pag 10: “Isolated fWJ-MSCs confirmed the possibility of culturing them through several dozen passages.” This sentence is not appropriate because you have not investigated if some markers can change during the in vitro culture that could induce some epigenetic changes
Author Response
The paper of Seo et al. is interesting because study an interesting stem cells source in the feline species.
The paper is well written and well organized with important experimental studies to define the characteristics of these stem cells. I have only some suggestions:
To define the characteristics of a stem cell line, it is very important to check the modification of these cells during the different passages in vitro. So, it would have been important to check the positivity to the mesenchymal markers and the negativity to the hematopoietic markers at different passages: for example, at passage 0, 5 and 10 ....
â–¶ Thank you for your kindly comments. I agree with the reviewer's comment. The purpose of this study is to confirm the isolation, culture and characterization of fWJ-MSCs. We recognize that the reviewer's comment is a part that must be confirmed optimal passage for clinical trial with fWJ-MSCs. After this study, when attempting clinical trial using fWJ-MSCs, we plan to proceed with the study on the changes of characteristic on the passage mentioned above. Also, please consider the short review period (7 days).
Moreover, I do not understand why the flow cytometry and differentiation studies were done at the step 5 and the gene expression studies through PCR at the step 3.
â–¶ Thank you for your kindly comments. That is a mistake. All characterization experiments were performed in passage 5, modified the materials and methods (noted by red color; p2 line 80).
At last, we know that this source is heterologous so, in the case of therapy these cells will be used cryopreserved, so it would have been important to understand if these cells tolerate the cryopreservation and if after cryopreservation they maintain the same proliferative, differentiative properties and the expression of stem cell markers.
â–¶ Thank you for your kindly comments. I agree with the reviewer's comment. We recognize that the reviewer's comment is a part that must be confirmed influences of cryopreservation for clinical trial with fWJ-MSCs. In this study, fWJ-MSCs were isolated and stocked in passage 3 and stored in liquid nitrogen for one month. Characterization experiments were performed at passage 5 after thawing, but the comparison with cells without cryopreservation was not performed. So it is thought that the torelance for cryopreservation within passage 5 is not large, but further research is needed. Further studies will be proceeded on influences of cryopreservation for clinical trial study. Also, please consider the short review period (7 days).
This paper need to some improvement about these issues
In addition, there are some suggestions:
Line 33-35 pag 1: please change “MSCs were proposed as an important option in the field of regenerative medicine and immunotherapy due to its immunosuppressive properties, its ability to produce cytokines or growth factors, and its ability to differentiate multilineage such…. “ In “MSCs were proposed as an important option in the field of regenerative medicine and immunotherapy due to their immunosuppressive properties, their ability to produce cytokines or growth factors, and their ability to differentiate multilineage such…”
â–¶ Thank you for your kindly comments. Modified as you have given feedback (noted by red color; p1 line 33-36).
Line 41 pag 1 “… cow, mouse (10-15).”Here are reported only citations about the isolation from BM but there are many papers in veterinary field about the isolation of cells from other sources as for example, amniotic membrane or amnioric fluid or wharthon jelly....please insert some papers about this (amniotic membrane, amniotic fluid and wharthon jelly were studied in many animal species by Lange-Consiglio et al.).
â–¶ Thank you for your kindly comments. I agree with the reviewer's comment. Modified as you have given feedback (noted by red color; p11 361-363, 366-372)
Line 72 pag 2: “by centrifugation at 3,000 rpm for 5 min.” please change with “g” that is an universal value
â–¶ Thank you for your kindly comments. Modified as you have given feedback (noted by red color; p2 line 73).
Line 80 pag 2: “Total RNA was….” Please insert here at what passage you have performed this study
â–¶ Thank you for your kindly comments. Added as you have given feedback (noted by red color; p2 line 80).
Line 96 pag 3: “…levels of the previous passages….” Please insert here until to which passage you have performed this study
â–¶ Thank you for your kindly comments. Added as you have given feedback (noted by red color; p3 line 96).
Line 110 pag 3: “…at 1,000 rpm for 10…” please insert “g” and not rpm
â–¶ Thank you for your kindly comments. Modified as you have given feedback (noted by red color; p3 line 110).
Line 111 pag 3: “….at 37°C. And the cells…” please change in “…at 37°C, then the cells...”
â–¶ Thank you for your kindly comments. Modified as you have given feedback (noted by red color; p3 line 111).
Line 129 pag 4: “Osteogenic differentiation medium…..” please, for each differentiation study insert that you have performed the negative control
â–¶ Thank you for your kindly comments. Added as you have given feedback (noted by red color; p4 line 132-133, 150, 170-171).
Line 290 pag 9: “In previous study showed that…” please change in “Previous study showed that….”
â–¶ Thank you for your kindly comments. Modified as you have given feedback (noted by red color; p9 line 294).
Line 295 pag 9: “WJ-MSCs…” please, do not use the acronym at the at the beginning of a sentence. Please check through all the manuscript
â–¶ Thank you for your kindly comments. Modified as you have given feedback (noted by red color; p9 line 294, 297).
Line 320 pag 10: “Isolated fWJ-MSCs confirmed the possibility of culturing them through several dozen passages.” This sentence is not appropriate because you have not investigated if some markers can change during the in vitro culture that could induce some epigenetic changes
â–¶ Thank you for your kindly comments. I agree with the reviewer's comment. Deleted as you have given feedback.
Round 2
Reviewer 4 Report
The paper is improved. I suggest the authors to check through all manuscripts and legends that the name of genes are written in italics
Author Response
The paper is improved. I suggest the authors to check through all manuscripts and legends that the name of genes are written in italics
â–¶ Thank you for your kindly comments. Modified as you have given feedback (noted by red color; p3 table 2, p5 line 182, p5 line 205, p6 line 212~213, p6 line 222~225 p6 figure 2B, p7 line 237, p7 line 247, p7 line 254, p8 line 264, p8 line 273, p9 line 281, p9 line 298~299, p10 line 310~313).
